# A Review of Fish Vaccine Development Strategies: Conventional Methods and Modern Biotechnological Approaches

**DOI:** 10.3390/microorganisms7110569

**Published:** 2019-11-16

**Authors:** Jie Ma, Timothy J. Bruce, Evan M. Jones, Kenneth D. Cain

**Affiliations:** 1Department of Fish and Wildlife Sciences, College of Natural Resources, University of Idaho, Moscow, ID 83844, USAtbruce@uidaho.edu (T.J.B.); evanj@uidaho.edu (E.M.J.); 2Aquaculture Research Institute, University of Idaho, Moscow, ID 83844, USA

**Keywords:** aquaculture, conventional vaccines, alternative vaccine, technologies

## Abstract

Fish immunization has been carried out for over 50 years and is generally accepted as an effective method for preventing a wide range of bacterial and viral diseases. Vaccination efforts contribute to environmental, social, and economic sustainability in global aquaculture. Most licensed fish vaccines have traditionally been inactivated microorganisms that were formulated with adjuvants and delivered through immersion or injection routes. Live vaccines are more efficacious, as they mimic natural pathogen infection and generate a strong antibody response, thus having a greater potential to be administered via oral or immersion routes. Modern vaccine technology has targeted specific pathogen components, and vaccines developed using such approaches may include subunit, or recombinant, DNA/RNA particle vaccines. These advanced technologies have been developed globally and appear to induce greater levels of immunity than traditional fish vaccines. Advanced technologies have shown great promise for the future of aquaculture vaccines and will provide health benefits and enhanced economic potential for producers. This review describes the use of conventional aquaculture vaccines and provides an overview of current molecular approaches and strategies that are promising for new aquaculture vaccine development.

## 1. Introduction

Despite multiple approaches to innovative therapy, fish diseases remain a major economic issue in commercial aquaculture worldwide. Although antibiotics or chemotherapeutics may be implemented for disease treatment, there are some clear drawbacks, such as drug resistance issues and safety concerns [1]. Vaccination, as an effective method of preventing a wide range of bacterial and viral diseases, and contributes to environmental, social, and economic sustainability in global aquaculture. Since the first reports in the 1940s of fish vaccination for disease prevention [2], there have been many vaccines developed that significantly reduced the impact of bacterial and some viral diseases in fish [3]. Millions of fish are vaccinated annually, and in some areas of the world there has been a transition away from antibiotics and toward vaccination. For example, there has been a dramatic reduction in the use of antibiotics in Norwegian salmon farming since the introduction of vaccines [4], and vaccination has become the most cost-effective and sustainable method of controlling infectious fish diseases [5].

A typical fish vaccine either contains or produces a substance that serves as an antigen. This component then stimulates an innate and/or adaptive immune response within the fish against a particular pathogen. Research on fish vaccines and fish immunology has increased throughout the 20th century, and there have been over 10,000 scholarly publications on fish vaccines in just the past 20 years. Several review articles described the history, advancements, types, and administration routes of fish vaccines, as well as the prospects and challenges of developing vaccines in aquaculture [6,7]. A historical review on fish vaccine research and the early pioneers in this field was published by Gudding and his colleagues [8]. The use of adjuvants and immunostimulants in fish vaccines, along with delivery methods, has been summarized by many researchers. Such work has focused on alternative methods (other than injection) for vaccine delivery, and the protective efficacies of traditional and promising new generation adjuvants [9,10,11]. Sommerset et al. described commercially available fish vaccines and how they perceived that the field would evolve over time [12]. Additional reviews have focused on current vaccine applications for large-scale fish farming operations, and future trends for inactivated, live-attenuated, and DNA vaccines [13]. Given the development of new technology and a lack of research reviews on advancements in fish vaccine technologies, there is a need for a comprehensive overview of where the field is currently. Up to now, over 26 licensed fish vaccines are commercially available worldwide for use in a variety of fish species (Table 1). Most of the vaccines have been approved for use by the United States Department of Agriculture (USDA) for a variety of aquaculture species, and the majority of these vaccines utilize conventional production methods that start by culturing target pathogens [14,15]. This array of vaccines has successfully protected fish against a variety of serious fish diseases.

A strong understanding in fish vaccinology is generally based on two sciences: microbiology and immunology. With the advancement of molecular biology and the improved knowledge of protective antigens, there are rapid developments for new generations of vaccines for use in animals and humans [16,17,18,19]. Modern vaccine technology has targeted specific pathogen components, and vaccines developed using such approaches may include subunit or recombinant DNA vaccines that contain novel antigens produced using various expression systems [19,20]. Other technologies, such as RNA particle vaccines, have been developed globally and appear to induce greater levels of immunity than traditional vaccine technology [18]. Overall, such advances are promising; however, actual implementation had been somewhat limited for aquaculture due in part to the challenges of the aqueous environment and practical application of mass vaccination due to the nature of commercial fish farming [21]. In this review, we describe the use of conventional aquaculture vaccines and provide an overview of molecular approaches to vaccine development that are the most promising for new vaccines for use in aquaculture.

## 2. Conventional Fish Vaccines

Conventional fish vaccines have primarily consisted of inactivated whole organisms, but some live attenuated or subunit protein vaccines (formulated with adjuvants) have been commercialized [26]. The majority of licensed vaccines presently used in aquaculture are produced using conventional methods and principles similar to those initially developed by Jenner and Pasteur centuries ago [27]. Early fish vaccines consisted of formalin-killed bacteria, with or without adjuvant [10,11]. These were delivered through immersion or injection routes and in turn induced some level of humoral immunity (Figure 1). In the 1990s, some modified live vaccines were developed and commercialized for use in aquaculture [28]. These vaccines have been successful and their implementation has resulted in increased production for commercial aquaculture along with reduced use of chemical therapeutics and feed delivered antibiotics [3,8].

### 2.1. Inactivated/Killed Vaccine

Inactivated or killed vaccines are typically created from a virulent disease-causing microbe, and through some process it loses its ability to infect or replicate in or outside of a host. These changes can be induced through physical, chemical, or radiation processes without compromising the antigenicity of the microbial agent [29]. In contrast to live vaccines (discussed below) inactivated vaccines are more stable under field conditions and may be less expensive to produce [30]. Inactivated vaccines do not persist within the environment or in the vaccinated fish, so they are usually found to be safe, but may induce weaker or shorter-lived immunity when compared other vaccine types [31]. Weak immunogenicity of inactivated vaccines may be attributed to a poor activation of cellular immunity within the fish species and, therefore, can necessitate the use of adjuvants or multiple booster immunizations to induce protective immunity. Once administered, phagocytic antigen presenting cells (APCs) begin the process of removing activated immune cells and eliciting a humoral immune response with memory. Disadvantages for inactivated vaccines include the potential for immunosuppressive passenger antigens, toxic reactions caused by immune-enhancing adjuvants, reduced immunogenicity due to denaturation of proteins, and systemic reactions to certain adjuvants [32].

In aquaculture, most early vaccine trials focused on killed vaccines. The first reported use of a fish vaccine was a killed *Aeromonas salmonicida* vaccine by Duff, who investigated oral vaccination of cutthroat trout *Oncorhynchus clarkii*. The first commercially licensed vaccine for fish was a killed *Yersinia ruckeri* vaccine delivered by immersion against enteric redmouth disease [3]. Following the success of this bacterin, formalin-killed immersion vaccines for vibriosis (caused by *Vibrio* spp.) of trout and salmon were developed. The same principle for the inactivation of bacterial pathogens of Atlantic salmon (*Salmo salar*) was used to develop the early salmonid vaccines that were delivered by immersion [28]. These early immersion vaccines against *A. salmonicida* were not effective, as Bricknell et al. reported on the first injection-based bacterial vaccine in Atlantic salmon [33]. Currently, large-scale commercial aquaculture operations, especially those focused on high-value species such as Atlantic salmon, primarily utilize killed polyvalent injectable vaccines that contain adjuvant and multiple antigens to protect against different diseases [12,34]. Four of the eight licensed vaccines for fish in the US are killed vaccines (Table 1). These include: an *A. salmonicida* bacterin for use in salmonids and Koi carp (*Cyprinus carpio*), an *A. salmonicida-Vibro anguillarum-Vibrio ordalii-Vibrio salmonicida* bacterin for use in salmonids, an infectious salmon anemia (ISA) virus vaccine, and a *Y. ruckeri* bacterin for use in salmonids. Killed vaccines against *Streptococcus* spp. or/and *Lactococcus* spp. infections in rainbow trout (*Oncorhynchus mykiss*) or amberjack (*Seriola dumerili*) and yellowtail (*Seriola quinqueradiata*) in Japan and Europe also exist [12]. There is also a killed *vibriosis* vaccine, combined with *Photobacterium damselae* (subsp. *piscicida*), available and used in European sea bass and sea bream culture (Table 1). So far, there is only one bacterin (ERM vaccine) commercially available as an oral vaccine. Autogenous vaccines are created from site-specific, isolated pathogens of interest and provide more flexibility in production regulation, and are implemented within a collaborative veterinary-client-patient-relationship [35]. Inactivated autogenous homologous vaccines, which are often killed bacterial strains, may provide producers with a cost-effective alternative to commercial vaccines, and these vaccines can be catered to specific pathogens that are problematic to a particular operation [6,14]. These vaccines may offer a responsive solution to emerging pathogens of interest, where commercial vaccines may not be applicable [35].

Inactivated vaccines are often less efficient against viral infections and diseases caused by intracellular bacteria [36]. A formalin-inactivated *Edwardsiella ictaluri* vaccine previously used in aquaculture was shown to have a limited ability to enter the fish [37]. Some studies have shown that inactivated vaccines do not generate sufficient immunity for salmon pancreas disease virus (SPDV) or red sea bream iridovirus disease [38]. Additionally, it is difficult to obtain long-lasting immunity against salmon rickettsial syndrome (*Piscirickettsia salmonis* infection). Booster immunizations can enhance immunity and even oral boosters have been shown to strengthen the immune response to a specific pathogen; for example, this was demonstrated for Francisellosis (*Franscisella* spp), which is caused by infection with *F. noatunensis* [39].

### 2.2. Live Vaccines

Modified live vaccines are prepared from one or more viruses or bacteria displaying attenuated virulence or natural low virulence toward the target fish species. Pathogens can be attenuated using physical or chemical processes, serial passage in cell culture, culture under abnormal conditions, or genetic manipulation [40]. Live vaccines tend to be more immunogenic than killed preparations due to their ability to proliferate or enter the host and stimulate greater cellular responses linked to both innate and adaptive immunity [41]. Such cell-mediated immune responses are considered to mimic a natural pathogen infection almost identically and, in turn, generate a strong antibody response. Since the pathogen enters and often replicates within the host, the animal can develop adequate cellular memory resulting in long-lasting immunity, which is clearly a major benefit in agricultural and aquaculture species [28]. Since live vaccines are usually quite effective and retain attributes of natural infection, the use of an accompanying adjuvant is not typically required to enhance efficacy. In terms of commercial applications for aquaculture, live vaccines have a greater potential to be administered via oral or immersion routes (Figure 1). Thus, the mode of administration is more dynamic than for inactivated vaccines that must utilize adjuvants [29].

Attenuated live vaccines have been proven safe under most circumstances; however, there are potential risks that must be addressed to ensure such products do not revert to virulence, display residual virulence, or are virulent in immunocompromised vaccinates. This, or the potential of contamination with unwanted organisms, could affect the efficacy and the licensing process for live vaccines. Presently, three modified live aquaculture vaccines are licensed in the USA. These include an *Arthrobacter* vaccine against bacterial kidney disease (BKD) for use in salmonids, an *E. ictalurii* vaccine against enteric septicemia of catfish (ESC), and a *Flavobacterium columnare* vaccine against columnaris in catfish [42]. The live *Arthrobacter* vaccine, named Renogen against BKD, has been licensed in Canada and Chile [28] and consists of non-pathogenic soil bacteria that provide cross-protection to the virulent *Renibacterium salmoninarum.* A live viral hemorrhagic septicemia virus (VHSV) vaccine is available in Germany [43], and a live viral vaccine against Koi herpesvirus (KHV) for carp is available for use in Israel [44].

The two licensed live bacterial vaccines in the US were developed by a serial passage procedure in the presence of increasing concentrations of the antibiotic rifampicin [25,45,46,47]. This strategy is one of the most successful chemical mutagenesis strategies for Gram-negative bacteria. Norqvist et al. utilized this approach for attenuation of *V. anguillarum* for use with rainbow trout [48]. This approach has also been used to attenuate *Flavobacterium psychrophilum*, and has demonstrated both safety and efficacy in salmonids [49]. Over the past decade, this *F. psychrophilum* vaccine has been further investigated and enhanced following production under iron limited conditions (designated B.17:ILM), has been optimized under a range of conditions, and has been shown to provide robust cross-protection against a wide variety of global *F. psychrophilum* strains [49,50,51,52,53,54,55]. This antibiotic mutagenesis approach was also implemented for other fish pathogens that include *A. hydrophila* [56] and *Flavobacterium* spp. [57,58,59] in channel catfish (*Ictalurus punctatus*) and Nile tilapia (*Oreochromis niloticus*) or carp species [60], *E. tarda* in channel catfish and Japanese flounder [61,62], and *V. anguillarum* in Japanese flounder [63]. Other chemical agents, like acriflavine dye and novobiocin, were used to attenuate *Streptococcus agalactiae*, *S. iniae, E. ictaluri,* and *A. hydrophila* [64,65], but, to date, these vaccines have not been commercialized.

The *Arthrobacter* vaccine against BKD is unique in that it does not consist of a live Gram-positive *R. salmoninarum* strain, but instead utilizes a live *A. davidanieli* bacterium that elicits cross-protective immunity to *R. salmoninarum* [24,66]. This use of this antigenic cross-reactivity microbe as an antigen can be effective, and other strategies to attenuate fish pathogens have been explored. These include serial passages [67] and the use of phylogenetic relatedness and antigenic cross-reactivity microbes as an antigen [24,66,68]. Physical processes were also used to induce mutations. For example, Perelberg et al. mutated an attenuated KHV by ultraviolet (UV) irradiation, which likely modified several additional viral genes, in an attempt to enhance attenuation and reduce the possibility of reversion back to a pathogenic phenotype [69].

Beyond physical and chemical induction of attenuation, molecular manipulation of pathogens has been implemented to produce genetically modified mutants as vaccine candidates. In most cases, the goal is to delete virulence genes or regulatory genes linked to virulence. This has been used for large DNA viruses, such as herpesviruses like KHV [70]. By selecting KHV mutants or KHV recombinants that targeted viral ribonucleotide reductase, thymidine kinase (TK), or dUTPase genes, it was found that similar levels of specific serum antibodies to the parental wild-type virus were induced, and it was suggested that these may serve as suitable live vaccines [44,71].

For bacterial strains, this technology has been successful in generating live vaccines against specific fish diseases. Some studies using genetic recombination that affected external polysaccharides of *Edwardsiella* spp. have been promising. By targeting the O-polysaccharides gene (OPS), polysaccharide biosynthesis was completely disrupted resulting in marked attenuation and high immune protection for catfish following challenge with the virulent wild-type bacterium [72,73,74,75,76,77,78]. Other studies have created attenuated *Streptococcus* spp., where virulence factor production, such as polysaccharides, M-like proteins, and phosphoglucomutase, was eliminated [79,80]. Other examples where genetic manipulation has resulted in attenuation include mutant *Francisella asiatica* [81], *Vibrio mimicus* [82], and *V. alginolyticus* strains [83]. These have provided protection in fish and suggested further development for live vaccine technologies for aquaculture.

## 3. Alternative Vaccine Technology

### 3.1. Subunit Vaccines

Subunit vaccines take advantage of using only antigenic components for vaccination and since subunit vaccines cannot replicate in the host, there is no risk of pathogenicity to the host or non-target species [84]. Subunit vaccines can be produced in a highly characterized state, and they target immune responses toward specific microbial determinants, enable the incorporation of unnatural components, and can be freeze-dried, allowing for non-refrigerated transport and storage [84,85]. Subunit vaccines have many desirable qualities, but in many cases, their ability to stimulate a potent immune response can be weaker than killed or live whole cell preparations. This is due to the limited number of components that are represented and capable of stimulating an immune system and the lack of replication or exposure to multiple antigens representing a whole cell vaccine. Some subunit vaccines rely on effective adjuvants to elicit the appropriate immunity, since the simplified (synthetic, recombinant, and/or highly purified) antigenic components of the vaccine mostly lack immunogenicity by themselves, and may require multiple booster immunizations to ensure long- term protective immunity [85].

There are multiple ways to produce a subunit vaccine. Immunogenic components can be isolated and purified directly from the target pathogen, or specific immunogenic proteins can be produced using various recombinant expression vectors. Different prokaryotic and eukaryotic cellular systems have been used for the production of immunogenic proteins for viral and bacterial fish vaccines. An *Escherichia coli* expression system has commonly been used for the propagation of plasmids carrying genes that encode a specific protective antigen, which can be harvested at the end of the fermentation cycle. Reverse vaccinology was adopted to design a unique multi-epitope subunit vaccine against pathogens. Based on this strategy, a four component, protein-based subunit vaccine against *Neisseria meningitidis* serogroup B vaccine was licensed for use by the European Medical Agency and other authorities [19,86].

Successful subunit vaccines for aquaculture that utilize *E. coli*-based expression include infectious pancreatic necrosis (IPN) in Norway (IPNV peptide, VP2; Produced by Merck Animal Health, Kenilworth, NJ, USA). A yeast-based subunit vaccine against the ISA virus (HE and F proteins; Manufactured by Centrovet) has also been produced and is available in Chile. Other expression systems used to produce the fish subunit vaccine experimentally include baculovirus and yeast for VHSV or infectious hematopoietic necrosis virus (IHNV) proteins [30,87,88,89,90,91], IPN proteins manufactured via cabbage worms (*Trichoplusia ni*) [92], and salmonid alphavirus 3 recombinant proteins produced in a fish cell line [93]. Although substantial research has been conducted on subunit vaccines, they have not been widely developed or commercialized for use in aquaculture. This is likely due to the need for adjuvants to stimulate adequate protection against some pathogens of concern [9,85,94,95,96]. Further, with respect to production costs, recombinant vaccines may often be expensive to produce for fish species, animals that may be considered low-value in comparison to other agricultural production species [97]. This is due to additional protein processing methods that are required to ensure vaccine efficacy [97]. Despite these limitations, this strategy may still be important and has the potential to succeed for pathogens that are difficult to cultivate, such as the piscine myocarditis virus (causing cardiomyopathy syndrome in Atlantic salmon).

Virus-like particles (VLPs) are components of an advanced subunit vaccine (Figure 1), which are formed from the self-assembly of viral capsid proteins into particles that mimic the natural structure of the virus [98]. However, unlike the actual viral particles, VLPs lack genomic material, precluding any possibility of reversion mutations or pathogenic infection [99]. VLPs are unable to replicate in the recipient, but can potentiate both innate and adaptive immune responses through the recognition of repetitive subunits and by producing high cellular and humoral responses [100]. VLPs, both non-enveloped and enveloped, have been produced in bacteria, yeast, transgenic plants, insect, mammalian, and cell-free platforms. Additionally, vaccine antigens can be produced as genetic fusions or chemical conjugates to viral structural proteins, resulting in chimeric VLPs [101].

Due to the advantages compared to other vaccine types, the interest in VLP technology has increased in recent years. Some highly purified VLP vaccines have been licensed and commercialized in humans, such as GlaxoSmithKline (GSK)’s Engerix (hepatitis B virus (HBV)), Cervarix (human papillomavirus (HPV)), and Merck’s Recombivax HB (HBV) and Gardasil (HPV) [102]. Recently, various VLP vaccine candidates for fish diseases have been developed. For example, nervous necrosis virus (NNV) VLP vaccines that utilized *E. coli*, yeast, baculovirus, and plant or cell-free self-assembled expression were produced. Those studies showed that VLPs with similar size and geometry to the native virus could elicit an antibody response in fish by injection [103,104,105,106,107,108]. Chien et al. and Cho et al. developed oral VLP vaccines against grouper NNV [109,110]. Dhar et al. demonstrated that the IPNV capsid protein VP2, expressed in yeast, self-assembles into subviral particles (SVPs) and the injection of the SVPs into rainbow trout elicits an immune response [111]. Guo et al. reported that two IHNV recombinant viruses displaying IPNV VP2 protein were generated using the RNA polymerase system against both IHNV and IPNV infection [112]. Based on these and other studies, VLPs have been shown to elicit strong immunogenicity and constitute a safe alternative to inactivated or attenuated vaccines.

### 3.2. Nucleic Acid Vaccines

Several nucleic acid vaccines have been developed for use in aquaculture over the past 20 years [113]. It has been suggested that these vaccines have the combined positive attributes of both live attenuated and subunit vaccines [113,114]. Nucleic acid vaccines consist of DNA or RNA encoding the antigen(s) of interest and are considered relatively simple to generate and safe to administer since they cannot revert to a pathogenic state [26]. Nucleotide-based vaccines have advantages over other vaccines that include production flexibility that is scalable and cost competitive, and the need for cold storage is eliminated. Newer RNA-based vaccines offer all the advantages of nucleotide-based approaches, including enhanced immunogenicity, and it has been suggested that this advanced technology may start a revolution in medicine.

#### 3.2.1. DNA Vaccines

DNA vaccines consist of an expression plasmid that carries a specific gene that codes for a selected antigenic protein, which when expressed in the host is expected to elicit a strong immune response. Plasmid production is scaled-up within bacterial cells, and the gene of interest is flanked by promoter and termination elements that facilitate expression within eukaryotic cells [114]. DNA vaccines are able to strongly activate cellular and humoral immunity. The development of DNA vaccines can be rapid and relatively straightforward if a protective antigen is known. DNA vaccines are often more effective in protecting against viral infections, and have been especially efficient against fish rhabdoviruses, as they usually utilize the same cellular mechanics that a virus utilizes once they enter a host cell [115]. DNA vaccines also have been efficacious in the prevention of fish exposure to intracellular bacteria, like *Mycobacterium marinum* [116,117].

The first reported DNA vaccine for use in aquaculture was against IHN and it was tested in rainbow trout [114]. In the past decade, other experimental DNA vaccines have been developed against a variety of aquatic pathogens and for a wide array of fish species [118]. However, only a limited number of these have been commercialized and made available for the market. One is a DNA vaccine against IHNV, which was licensed and commercialized in Canada (Apex-IHN), and another is against pancreas disease virus in the European Union (salmonid alphavirus subtype 3 DNA vaccine) and is marketed as Clynav (Table 1).

Several DNA vaccine reviews were published that report on their effects and application [114,115,119]. The route of administration for these vaccines is intramuscular (IM) injection in most fish species, as the genetic material must be relatively protected in order to gain entry into host cells [120]. After being injected with a DNA vaccine, fish demonstrate enhanced innate and adaptive immunity similar to mammalian species [115]. Additionally, DNA vaccines can usually be constructed to be multivalent, and provide protection or cross-protection by using gene coding for multiple antigens in the plasmid design [121]. DNA vaccines are considered safer than attenuated live vaccines as they only express the antigenic protein segments and not the entire organism, although the antigen interactions within the host are not well understood. As the antigen is produced inside of the organism via genetic expression from host cells, the duration of the immune response is in most cases long-lasting [122].

#### 3.2.2. RNA-Based Vaccines

At present, there are two major RNA-based vaccines, which are distinguished by the translational capacity of the RNA: conventional, non-amplifying mRNA and self-amplifying mRNA (i.e., replicons) [110]. The RNA-based vaccines are developing rapidly and several have demonstrated encouraging results in both human and animals [123]. Using RNA in a vaccine has a number of advantages: it is safe because RNA is non-infectious and degraded by normal cellular processes, and there is no potential risk of infection or insertional mutagenesis. Furthermore, RNA is a potent stimulator of immunity [123]. Recent advances in RNA vaccine technology have been extensively reviewed in several publications [26,123,124,125,126], and thus, the successes and impact are briefly summarized in the context of their future promise for application in aquaculture.

The most currently used self-replicating RNA vaccines are based on an alphavirus genome [127]. Alphaviruses belong to the family *Togaviridae,* which includes the Sindbis virus, Semliki Forest virus, and equine encephalitis viruses [128,129]. The alphavirus vector vaccine has a single RNA gene encoding RNA replication machinery, which is left intact, but the genes encoding the structural proteins are replaced with the antigen of interest. This antigen-encoding RNA replicon platform enables a large number of antigens from an extremely small dose of vaccine to be produced in vitro transcribed (IVT) from a DNA template [123]. The viral replication takes place in the cytoplasm of the host cell and, therefore, is independent of the host replication system, making replicase-based nucleic acid vaccines a very efficient and attractive delivery vehicle [124]. Previous research has demonstrated that alphaviral RNA vaccines are more efficient at stimulating antigen specific immune responses, particularly cellular responses, as compared with conventional plasmid DNA vaccines [130].

Alphavirus replicon particles (RPs) are single-cycle, propagation-defective particles that are not able to spread beyond the initial infected cells. RP vaccines have been evaluated in many different species of animals as well as humans with a proven record of safety and efficacy [131]. The RPs RNA vaccine that was approved by the United State Department of Agriculture (USDA) against influenza A virus in swine was produced using a unique SirraVaxSM RNA particle technology platform (Harris Vaccine Inc., now Merck Animal Health) [131,132]. In 2014, just four months after the porcine epidemic diarrhea virus (PEDV) broke out in the United States and affected about 50% of the swine population, Harris vaccines commercialized an RP vaccine that prevented this disease [133] The alphavirus RPs RNA vaccine has been developed and has been shown to provide protection against many pathogens in many different species [131,134,135,136]. These vaccines can induce robust and balanced immune responses and offer many other advantages associated with ideal vaccines. The alphavirus replicase functions in a broad range of host cells, like mammalian, avian, reptilian, amphibian, insect, and fish. Therefore, by replacing the genes for the structural proteins of the virus with a fish pathogen antigen of interest, the self-amplifying RNA vaccine could potentially protect against a number of important fish diseases. SPDV, a known salmonid alphavirus (SAV), was assigned to the family *Togaviridae*, genus *Alphavirus*, and resembles the genome of mammalian alphaviruses. Kalsen et al. described the characterization of untranslated regions of the salmonid alphavirus 3 (SAV3) genome and constructed an SAV3 based replicon [137]. This replicon vaccine provides high protection against ISA by IM injection without adjuvant, but intraperitoneal (IP) administration of the same replicon vaccine did not induce protection [138,139]. Hence, the SAV-based replicon represents a vaccine candidate for aquaculture.

## 4. Conclusion

An ideal fish vaccine is one that is safe for the animal and environment, economical for large-scale production, easy to administer, capable of inducing strong immunity throughout periods of greatest susceptibility, and demonstrates minimal side effects. New and alternative fish vaccines are adapting advanced technologies often developed based on needs in animal or human medicine, but have shown great promise for aquaculture. Those that meet the criteria for an effective aquaculture vaccine will provide the most benefit and have the greatest potential for commercialization. New fish vaccines using alternative technologies (beyond just killed cellular preparations) can be expensive to develop, but given the limited success of traditional approaches for new diseases problems, it is essential to further explore such approaches. As aquaculture continues to grow globally, there will be a need for new vaccines long into the future, and the application of all available biotechnology towards solving emerging disease issues will be critical.

## Figures and Tables

**Figure 1 microorganisms-07-00569-f001:**
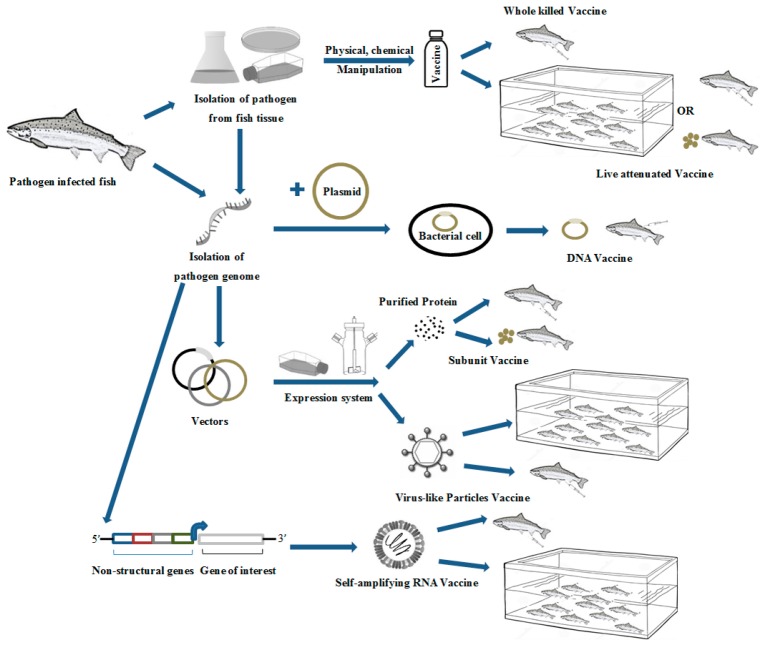
Various approaches for fish vaccine development. On the left side are preparatory components, followed by production means in the middle, and final administration routes to the right. Individual fish receiving an injection denote the injection vaccination, while fish consuming the feed pellets represent administration via oral uptake. The grouped fish within the tank represent immersion as a vaccination route.

**Table 1 microorganisms-07-00569-t001:** Overview of licensed fish vaccines that have been used in global aquaculture.

Disease	Pathogen	Major Fish Host	Vaccine Type	Antigens/Targets	Delivery Methods	Country/Region*	Further Information
**Viral Diseases**							
Infectious hematopoietic necrosis	IHNV *Rhabdovirus*	Salmonids	DNA	G Glycoprotein	IM	Canada	https://www.dfo-mpo.gc.ca/aquaculture/rp-pr/acrdp-pcrda/projects-projets/P-07-04-010-eng.html
Infectious pancreatic necrosis	IPNV *Birnavirus*	Salmonids, sea bass, sea bream, turbot, Pacific cod	Inactivated	Inactivated IPNV	IP	Norway, Chile, UK	www.pharmaq.no
			Subunit	VP2 and VP3 Capsid Proteins	Oral	Canada, USA	www.aquavac-vaccines.com
			Subunit	VP2 Proteins	IP	Canada, Chile, Norway	http://www.msd-animal-health.no/
Infectious salmon anemia	ISAV *Orthomyxovirus*	Atlantic salmon	Inactivated	Inactivated ISAV	IP	Norway, Chile, Ireland, Finland, Canada	www.pharmaq.no
Pancreatic disease virus	SAV *alphaviruses*	Salmonids	Inactivated	Inactivated SAV	IP	Norway, Chile, UK	https://www.merck-animal-health.co
Spring viremia of carp virus	SVCV *Rhabdovirus*	Carp	Subunit	G Glycoprotein	IP	Belgium	[22]
			Inactivated	Inactivated SVCV	IP	Czech Republic	[23]
Koi herpesvirus disease	KHV *Herpesvirus*	Carp	Attenuated	Attenuated KHV	IMM or IP	Israel	[22]
Infectious spleen and kidney necrosis	ISKNV *Iridovirus*	Asian seabass, grouper, Japanese yellowtail	Inactivated	Inactivated ISKNV	IP	Singapore	https://www.aquavac-vaccines.com/
**Bacterial diseases**							
Enteric redmouth disease (ERM)	*Yersinia ruckeri*	Salmonids	Inactivated	Inactivated *Y. ruckeri*	IMM or oral	USA, Canada, Europe	http://www.msd-animal-health.ie/products_ni_vet/aquavac-erm-oral/overview.aspx; https://www.msd-animal-health-hub.co.uk
Vibriosis	*Vibrio anguillarum;* *Vibrio ordalii;* *Vibrio salmonicida*	Salmonids, ayu, grouper, sea bass, sea bream, yellowtail, cod, halibut	Inactivated	Inactivated *Vibriosis* spp.	IP or IMM	USA, Canada, Japan, Europe, Australia	https://www.merck-animal-health.com/species/aquaculture/trout.aspx;
Furunculosis	*Aeromonas salmonicida* subsp. *salmonicida*	Salmonids	Inactivated	Inactivated *A. salmonicida* spp.	IP or IMM	USA, Canada, Chile, Europe, Australia	https://www.msd-animal-health-me.com/species/aqua.aspx
Bacterial kidney disease (BKD)	*Renibacterium salmoninarum*	Salmonids	Avirulent live culture	*Arthrobacter davidanieli*	IP	Canada, Chile, USA	[24]
Enteric septicemia of catfish (ESC)	*Edwarsiella ictaluri*	Catfish	Inactivated	Inactivated *E. ictaluri*	IP	Vietnam	https://www.pharmaq.no/
Columnaris disease	*Flavobacterium columnaris*	All freshwater finfish species, bream, bass, turbot, salmon	Attenuated	Attenuated *F. columnare*	IMM	USA	[25]
Pasteurellosis	*Pasteurela piscicida*	Sea bass, sea bream, sole	Inactivated	Inactivated *P. pscicida*	IMM	USA, Europe, Taiwan, Japan	ALPHA JECT 2000
Lactococciosis	*Lactococcus garviae*	Rainbow trout, amberjack, yellowtail	Inactivated	Inactivated *L. garviae*	IP	Spain	https://www.hipra.com/
Streptococcus infections	*Streptococcus* spp.	Tilapia, yellow tail, rainbow trout, ayu, sea bass, sea bream	Inactivated	Inactivated *S. agalactiae* (biotype 1)	IP	Taiwan Province of China, Japan, Brazil, Indonesia	https://www.aquavac-vaccines.com/products/aquavac-strep-sa1/
				Inactivated *S. agalactiae* (biotype 2)	IP		https://www.aquavac-vaccines.com/products/aquavac-strep-sa/
				Inactivated *S. iniae*	IP or IMM		https://www.aquavac-vaccines.com/products/aquavac-strep-si/
Salmonid rickettsial septicemia	*Piscirickettsia salmonis*	Salmonids	Inactivated	Inactivated *P. salmonis*	IP	Chile	Evensen, 2016; https://www.pharmaq.no/products/injectable/
Motile *Aeromonas* septicemia (MAS)	*Aeromonas* spp.	Striped catfish	Inactivated	*A. hydrophila* (serotype A and B)	IP	Vietnam	https://www.pharmaq.no/; ALPHAJECT Panga 2
Wound Disease	*Moritella viscosa*	Salmonids	Inactivated	Inactivated *M. viscosa*	IP	Norway, UK, Ireland, Iceland	https://www.pharmaq.no
Tenacibaculosis	*Tenacibaculum maritimum*	Turbot	Inactivated	Inactivated *T. maritimum*	IP	Spain	https://www.hipra.com/

IHNV: Infectious hematopoietic necrosis virus; IPNV: Infectious pancreatic necrosis virus; ISAV: Infectious salmon anemia virus; SVCV: Spring viremia of carp virus; KHV: Koi herpesvirus; ISKNV: Infectious spleen and kidney necrosis virus; IM: Intramuscular injection; IP: Intraperitoneal injection; IMM: Immersion; * denotes country or region where the vaccine is licensed and sold.

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
