# Peer review of "A Review of Fish Vaccine Development Strategies: Conventional Methods and Modern Biotechnological Approaches"

_microorganisms, 2019, doi:10.3390/microorganisms7110569_

Round 1
Reviewer 1 Report
This is a well-written review on fish vaccines that describes the state of the art very thoroughly. I have some comments that I hope will help the authors - none of them are major comments.
line 62-74: this section in the introduction is very focused on animal vaccines, and specifically on virus particles (ignoring other approaches used for e.g. human vaccines that have potential also for fish, maybe?). This section could have more information on recombinant vaccine strategies also for bacterial diseases, and on what e.g. Rino Rappuoli has termed "reverse vaccinology".
I also miss reference to the cost per dose of recombinant or molecular approaches compared to traditional, cheap 'dead' (boiled microbe) vaccines... this to my understanding is the greatest limit to the development of recombinant fish vaccines.
The matter of subunit vaccines (i.e. recombinant vaccines) is picked up later again (references 81-87... line 129ff) but again is exclusively discussed for existing fish approaches. I would wish for a bit more discussion of approaches that have not yet been tried in fish (but work in human)? Human virus particle vaccines are discussed in some detail (line 167ff) but not examples like the meningococcal (bacterial) vaccines
See https://www.ncbi.nlm.nih.gov/pubmed/16107075 for what I am referring to as an example.
Author Response
Thank you to the Microorganisms editorial staff and reviewers for providing comments on our manuscript. Our responses are given in a point-by-point manner below:
Review 1
This is a well-written review on fish vaccines that describes the state of the art very thoroughly. I have some comments that I hope will help the authors - none of them are major comments.
line 62-74: this section in the introduction is very focused on animal vaccines, and specifically on virus particles (ignoring other approaches used for e.g. human vaccines that have potential also for fish, maybe?). This section could have more information on recombinant vaccine strategies also for bacterial diseases, and on what e.g. Rino Rappuoli has termed "reverse vaccinology".
Thanks for the comments. Dominic F. Kelly and Rino Rappuoli (2005) paper gave a very focused review on reverse vaccinology, which relied on recombinant protein expression. So we added more details on reverse vaccinology within the subunit vaccine section, along with additional references.
I also miss reference to the cost per dose of recombinant or molecular approaches compared to traditional, cheap 'dead' (boiled microbe) vaccines... this to my understanding is the greatest limit to the development of recombinant fish vaccines.
Thank you, we have included an appropriate reference for this issue within the Subunit Vaccine section (line161-164). Barnes (2019) reports that the downstream processing for the recombinants can be expensive for fish, as they are low-value food species.
The matter of subunit vaccines (i.e. recombinant vaccines) is picked up later again (references 81-87... line 129ff) but again is exclusively discussed for existing fish approaches. I would wish for a bit more discussion of approaches that have not yet been tried in fish (but work in human)? Human virus particle vaccines are discussed in some detail (line 167ff) but not examples like the meningococcal (bacterial) vaccines
See https://www.ncbi.nlm.nih.gov/pubmed/16107075 for what I am referring to as an example.
Thanks. Addressed this section, as described above.
Reviewer 2 Report
This is a fairly well-written, focused manuscript on the variety of commercial vaccines available for aquaculture. The only major suggestion would be to include a short discussion of autogenous vaccines that can also be commercially available for specific pathogens.
Minor comments the authors’ should consider in a revision:
Page 1, line 35 – “areas” of what? Need clarification, i.e. “areas of the world”?
Page 2, line 56 – delete “at”. Do not end a sentence with a dangling preposition.
Page 6, line 43 – suggest placing salmonids before Koi carp because of vaccine’s importance to and use in this species.
Page 6, lines 48, 50 – suggest removing all references to specific companies in the body of the text since they are already mentioned in Table 1. Also, Merck is the only company you specifically mention by name which might imply a bias in article.
Page 7, line 78 – potential contamination affects BOTH the “efficacy” and licensing process.
Page 7, lines 90 and 102 - “gram” should be upper cased as it is a proper noun.
Page 7, line 90 –since you mention the species associated with the A. hydrophila and Flavobacterium spp vaccine, suggest mentioning the E. tarda vaccine with channel catfish and Japanese flounder, and the V. anguillarum vaccine with Japanese flounder for consistency.
Page 7, line 122 – delete “of”.
Page 8, line 137-139 – this statement of needing adjuvants with subunit vaccines is not true.
Page 8, line 143 – replace “of” with “for”.
Page 8, line 154 – again, this statement about adjuvants is not accurate.
Page 9, line 188 – delete “there”, as the word is not necessary in sentence.
Page 9, line 200 - you should also mention the DNA vaccine for mycobacterium which showed efficacy (Pasnik, D.J. and S.A. Smith. 2006. Diseases of Aquatic Organisms 73:33-41, and Pasnik D.J. and S.A. Smith. 2005. Veterinary Immunology and Immunopathology 103:195-206).
Page 9, line 215 – delete “more” as it is a relative term and unnecessary in sentence.
Page 9, line 223 – delete “extremely” as it is a relative term and unnecessary in sentence.
Page 12, lines 356-358 – different font size.
Page 13, lines 364 – different font size.
Figure 1 – very nice diagram, but not sure of the difference between “a single fish” vs. “a tank of fish” on right side of figure. Figure legend should also contain more text explaining figure.
Table 1 – There should be a dividing line between the IPN and ISA diseases, also there is an improperly extended line in the spring viremia of carp. Please clarify “Country/Region”, i.e. is this where the vaccine is approved or sold, or both. Don’t know why “immersion” is abbreviated “i.mm”, should just be “imm” without the abbreviating period.
Author Response
Thank you to the Microorganisms editorial staff and reviewers for providing comments on our manuscript. Our responses are given in a point-by-point manner below:
Review 2,
This is a fairly well-written, focused manuscript on the variety of commercial vaccines available for aquaculture. The only major suggestion would be to include a short discussion of autogenous vaccines that can also be commercially available for specific pathogens.
Thanks for the comments. Autogenous vaccines are created from site-specific, isolated pathogens of interest and provide more flexibility in production regulation, and are implemented within a collaborative veterinary-client-patient-relationship (VCPR; Yanong 2011). Autogenous vaccines, which are often killed bacterial strains, may provide producers with a cost-effective alternative to commercial vaccines, and these vaccines can be catered to specific pathogens that are problematic to a particular operation. So we added the autogenous vaccines in the killed vaccine section. These vaccines may offer a responsive solution to emerging pathogens of interest, where commercial vaccines may not be applicable (Adams 2019; Sudheesh and Cain 2017).
Minor comments the authors’ should consider in a revision:
Page 1, line 35 – “areas” of what? Need clarification, i.e. “areas of the world”?
Thank you, revised.
Page 2, line 56 – delete “at”. Do not end a sentence with a dangling preposition. Revised.
Page 6, line 43 – suggest placing salmonids before Koi carp because of vaccine’s importance to and use in this species. Revised.
Page 6, lines 48, 50 – suggest removing all references to specific companies in the body of the text since they are already mentioned in Table 1. Also, Merck is the only company you specifically mention by name which might imply a bias in article. Removed some companies in the text.
Page 7, line 78 – potential contamination affects BOTH the “efficacy” and licensing process. Revised.
Page 7, lines 90 and 102 - “gram” should be upper cased as it is a proper noun. Revised.
Page 7, line 90 –since you mention the species associated with the A. hydrophila and Flavobacterium spp vaccine, suggest mentioning the E. tarda vaccine with channel catfish and Japanese flounder, and the V. anguillarum vaccine with Japanese flounder for consistency. Revised.
Page 7, line 122 – delete “of”. Deleted.
Page 8, line 137-139 – this statement of needing adjuvants with subunit vaccines is not true.
We have made this revision (line 137-141), and added new reference.
Page 8, line 143 – replace “of” with “for”. Replaced.
Page 8, line 154 – again, this statement about adjuvants is not accurate. Addressed, as described above (Page 8, line 137).
Page 9, line 188 – delete “there”, as the word is not necessary in sentence. Deleted.
Page 9, line 200 - you should also mention the DNA vaccine for mycobacterium which showed efficacy (Pasnik, D.J. and S.A. Smith. 2006. Diseases of Aquatic Organisms 73:33-41, and Pasnik D.J. and S.A. Smith. 2005. Veterinary Immunology and Immunopathology 103:195-206).
Added this information and references in DNA vaccines section.
Page 9, line 215 – delete “more” as it is a relative term and unnecessary in sentence. Deleted.
Page 9, line 223 – delete “extremely” as it is a relative term and unnecessary in sentence. Deleted.
Page 12, lines 356-358 – different font size. Revised.
Page 13, lines 364 – different font size. Revised.
Figure 1 – very nice diagram, but not sure of the difference between “a single fish” vs. “a tank of fish” on right side of figure. Figure legend should also contain more text explaining figure.
“A single fish” and “a tank of fish” stood for different routes, oral or injection, and immersion routes, respectively. The figure legend was expanded to include these additional details.
Table 1 – There should be a dividing line between the IPN and ISA diseases, also there is an improperly extended line in the spring viremia of carp. Please clarify “Country/Region”, i.e. is this where the vaccine is approved or sold, or both. Don’t know why “immersion” is abbreviated “i.mm”, should just be “imm” without the abbreviating period. Thank you, the line was added into the figure and imm was adjusted and listed in the figure legend. Clarified that country/region refers to where the vaccines are licensed and sold.